# Established Liposome-Coated IMB16-4 Polymeric Nanoparticles (LNPs) for Increasing Cellular Uptake and Anti-Fibrotic Effects In Vitro

**DOI:** 10.3390/molecules27123738

**Published:** 2022-06-10

**Authors:** Xia Niu, Yanan Meng, Yucheng Wang, Guiling Li

**Affiliations:** Institute of Medicinal Biotechnology, Chinese Academy of Medical Science & Peking Union Medical College, Beijing 100050, China; niuxia@imb.pumc.edu.cn (X.N.); mengyanan@imb.pumc.edu.cn (Y.M.)

**Keywords:** liposome-coated polymeric nanoparticles, IMB16-4, liver fibrosis, hepatic stellate cells, cell uptake

## Abstract

As a global health problem, liver fibrosis still does not have approved treatment. It was proved that *N*-(3,4,5-trichlorophenyl)-2(3-nitrobenzenesulfonamide) benzamide (IMB16-4) has anti-hepatic fibrosis activity. However, IMB16-4 displays poor water solubility and poor bioavailability. We are devoted to developing biodegraded liposome-coated polymeric nanoparticles (LNPs) as IMB16-4 delivery systems for improving aqueous solubility, cellular uptake, and anti-fibrotic effects. The physical states of IMB16-4−LNPs were analyzed using a transmission electron microscope (TEM), high-performance liquid chromatography (HPLC), Fourier transform infrared spectroscopy (FTIR), X-ray diffraction (XRD) and differential scanning calorimeter (DSC). The results show that IMB16-4−LNPs increased the drug loading compared to liposomes and enhanced cellular uptake behavior compared with IMB16-4−NPs. In addition, IMB16-4−LNPs could repress the expression of hepatic fibrogenesis-associated proteins, indicating that IMB16-4−LNPs exhibited evident anti-fibrotic effects.

## 1. Introduction

Hepatic fibrosis, developed from chronic liver diseases, is considered one of the global health threats [1,2]. When hepatic parenchymal cells are stimulated by physical, chemical, and biological infections, adjacent hepatocytes, Kupffer cells and sinus endothelial cells secrete a variety of cytokines through paracrine action. These cytokines activate hepatic stellate cells (HSCs), which transform into myofibroblasts (MFC), proliferate, and synthesize the extracellular matrix (ECM) [3,4,5,6]. This destroys the inherent function and histological structure of the liver [7,8,9]. Activated HSCs are characterized by the overexpression of α-smooth muscle actin (α-SMA) as well as cytokines, such as transforming growth factor (TGF-β) [10]. Thus, a desired strategy for the treatment of liver fibrosis is warranted that can suppress the activation of HSCs or promote the apoptosis of activated HSCs [11,12,13,14,15,16,17,18,19,20].

Some types of hepatic fibrosis showed hardly any reversal by removal of the trigger of chronic inflammation [3]. Therefore, anti-fibrotic strategies will be needed to suppress the progression of liver fibrosis [21]. Despite recent advances in treatments for liver fibrosis in current clinical modalities, the therapeutic efficacy still needs improvement [22,23,24]. In response to this difficulty, ample research efforts have been made toward the resolution of liver fibrosis to improve patients’ quality of life and relieve their economic burden [25,26].

*N*-(3,4,5-trichlorophenyl)-2(3-nitrobenzenesulfonamido) benzamide, abbreviated as IMB16-4, is a potential therapeutic agent for the treatment of cholestatic hepatic fibrosis. However, IMB16-4 is almost insoluble in water and displays poor oral bioavailability [27,28]. We developed some nano-delivery systems to increase the anti-fibrotic efficacy, including mesoporous silica, liposomes, and polymeric nanoparticles. To explore the advantages of liposomes and polymeric nanoparticles, we intended to construct biodegraded liposome-coated polymeric nanoparticles (LNPs) as drug delivery systems, which were composed of a lipid bilayer as the outer sphere and polymeric nanoparticles as the inner core [29,30,31]. LNPs possess a higher drug loading and decrease drug leakage compared to liposomes [32,33]. In addition, the phospholipid bilayer shell imparts many therapeutic benefits, such as easy modification, high biocompatibility, and low immunogenicity [34].

Herein, we first established IMB16-4−LNPs and discussed their stability, drug release, cell uptake, and anti-fibrotic effect in vitro. IMB16-4−LNPs were systematically studied using TEM, XRD, DSC, and HPLC. The anti-fibrotic effects of IMB16-4−LNPs were also measured on the human HSCs line (LX-2) by testing α-SMA and MMP2 protein activity.

## 2. Results and Discussion

### 2.1. The Morphology and Particle Size

IMB16-4−NPs as hydrophilic cores were prepared for utilizing hydrophilic materials, PVPK30 by the anti-solvent method, and displayed a nearly monodispersed spherical shape with a size of 50~80 nm. IMB16-4−LNPs and blank liposomes were prepared by the film dispersion method, and the lipid membrane was dispersed in IMB16-4−NPs suspensions and distilled water, respectively. IMB16-4−LNPs with a size of 100~200 nm displayed lipid bilayer-coated IMB16-4−NPs and showed an excellent size distribution (PDI < 0.3) (Figure 1 and Table 1). The particle size and average Zeta potential of IMB16-4−LNPs were 119.0 ± 43.1 and −26.6 mV, respectively. The film-dispersed distilled water without IMB16-4−NPs formed the blank liposome, and most of the liposomes were in the shape of an oval with a size of 20~60 nm (Figure 1B,E).

### 2.2. Entrapment Efficiency (EE) and Drug Loading

The concentration of IMB16-4 was examined by HPLC. The EE of LNPs was found to be 46.1%. The IMB16-4 content was 1.03 mg/mL in LNPs suspension before freeze-drying. In addition, we tried to load IMB16-4 in the liposome membrane. However, the maximum drug loading was 52.2 μg/mL before freeze-drying; otherwise, IMB16-4 as white precipitate was precipitated at a concentration of more than 52.2 μg/mL during dispersing distilled water. Clearly, IMB16-4−LNPs greatly increased drug loading compared to liposomes.

### 2.3. Characterization of IMB16-4−LNPs

The intermolecular interaction between IMB16-4−NPs and the phospholipid bilayer usually leads to changes in FTIR spectra. The spectra of blank liposomes, IMB16-4−LNPs, IMB16-4−NPs, and IMB16-4 are shown in Figure 2. Pure IMB16-4 contained a C–H (unsaturated) stretching vibration at 3080~3200 cm^−1^ and NH stretching mode at 3375 cm^−1^, which were lost both in IMB16-4−NPs and IMB16-4−LNPs, and there appeared to be C–H aliphatic stretching at 2950 cm^−1^ and a CH_2_ symmetrical stretching vibration at 2857 cm^−1^. However, 1668 cm^−1^ in IMB16-4−LNPs vanished and was substituted by 1637 cm^−1^. The results indicate that the phospholipid bilayer and IMB16-4−NPs might be combined through physical interaction.

Small molecules presented a very sharp endothermic peak at the melting temperature during DSC analysis. As shown in Figure 3A, IMB16-4 showed a sharp endothermic peak at 253~258 °C, whereas no trace of an endothermic peak was observed for IMB16-4−NPs and IMB16-4−LNPs, indicating a decrease in the crystallinity of IMB16-4. Furthermore, we chose the XRD method to confirm the crystalline form. As shown in Figure 3B, the diffraction pattern of IMB16-4 was shown to be highly crystalline by the numerous peaks. However, no crystalline IMB16-4 was detected in IMB16-4−NPs or IMB16-4−LNPs, indicating that IMB16-4 existed as amorphous in IMB16-4−NPs and IMB16-4−LNPs.

### 2.4. Stability Measurements

IMB16-4 content, the size and Zeta potential of LNPs lyophilized powder were explored to measure its stability. Table 2 shows that the size, Zeta potential and IMB16-4 content of lyophilized powder did not significantly change during two months at 4 °C, indicating certain long-term storage stability.

### 2.5. In Vitro Release

The in vitro release behaviors of IMB16-4−LNPs and IMB16-4−NPs were conducted by the dialysis method in a 3% sodium dodecyl sulfate (SDS) release medium. As shown in Figure 4, IMB16-4 release from LNPs was obviously slower compared to that of IMB16-4−NPs at each time point. The release of IMB16-4 from IMB16-4−LNPs was evidently delayed before 12 h, at which time the accumulative dissolutions of IMB16-4−LNPs and IMB16-4−NPs were 17.8% and 37.4%, respectively, indicating that the release of IMB16-4−LNPs was somewhat controlled by lipid membrane.

### 2.6. In Vitro Cellular Uptake

The cellular uptake behavior of IMB16-4–LNPs and IMB16-4–NPs was quantitatively evaluated by HPLC under an equal concentration of protein. As illustrated in Table 3, IMB16-4–LNPs showed higher cellular uptake on the human HSCs line (LX-2) compared to IMB16-4–NPs, indicating that the lipid membrane increased the uptake of IMB16-4.

### 2.7. The Cytotoxicity

To examine the biocompatibility and cytotoxicity of LNPs formulation, the cytotoxicity of raw IMB16-4, IMB16-4−LNPs, IMB16-4−NPs, PVPK30, and blank liposomes at various concentrations was investigated in LX-2 cells (Figure 5). Blank liposomes and PVPK30 showed no cytotoxicity at the tested concentration. Raw IMB16-4 was dissolved in DMSO, and then diluted with DMEM/GlutaMAX I without FBS. Clearly, the cell survival rate of IMB16-4−LNPs at 40 μM was higher compared with that of raw IMB16-4. However, raw IMB16-4 was directly suspended in DMEM/GlutaMAX I and showed no cytotoxicity at the tested concentration, which might be attributed to a low water solubility.

The cell survival rate of IMB16-4−LNPs at 40 μM was lower compared to that of IMB16-4−NPs, which was caused by increased cellular uptake (Table 3). Interestingly, there was no obvious cytotoxicity of IMB16-4−LNPs at the concentrations from 2 μM to 10 μM, which enlarged the safe and effective concentration. The results showed that IMB16-4−LNPs reduced cytotoxicity and increased biocompatibility compared to raw IMB16-4. In addition, IMB16-4−LNPs reduced cytotoxicity at concentrations lower than 10 μM compared to IMB16-4−NPs, indicating IMB16-4−LNPs would be a suitable and pharmaceutically acceptable carrier.

### 2.8. In Vitro Antifibrotic Effects

LX-2 cells as human hepatic stellate cells were chosen to conduct anti-liver fibrosis effects in vitro. The TGF-β1 protein (2 ng/mL) further stimulated LX-2 cells into completely activated hepatic stellate cells. After being treated with TGF-β1, the hepaic fibrogenic makers from LX-2 were more secreted. Along with IMB16-4 treatment, the regulated expression of all hepaic fibrogenic makers is shown in Figure 6. IMB16-4−LNPs significantly decreased the protein levels of α-SMA and MMP2 on LX-2 cells. The anti-liver fibrosis effects of IMB16-4−LNPs were stronger than raw IMB16-4, indicating IMB16-4−LNPs increased anti-liver fibrosis effects. Furthermore, IMB16-4−LNPs better lowered the expression of MMP2 compared to IMB16-4−NPs, which can be attributed to the lipid membrane increasing its uptake of IMB16-4.

## 3. Materials and Methods

### 3.1. Materials

Egg yolk lecithin and cholesterol were purchased from AVT Pharmaceutical Tech Co., Ltd. (Shanghai, China). IMB16-4 was synthesized by our team (Lot No.: IMB20191213, Beijing, China). PVPK30 was purchased from Aladdin (Shanghai, China). Recombinant human TGF-β1 protein (TGF-β1) was purchased from R&D Systems (R&D, Minneapolis, Minnesota, USA). Antibodies for glyceraldehyde-3-antiphosphate dehydrogenase (GAPDH), MMP2, and α-SMA were obtained from Abcam (Abcam, Cambridge, UK). Horseradish peroxidase (HRP)-conjugated secondary antibodies against mouse or rabbit IgG were obtained from Proteintech (Wuhan, China). All other reagents were the class of reagent grade.

### 3.2. Preparation of IMB16-4−LNPs

IMB16-4−NPs were obtained as previously reported [27]. Then, 10 mg IMB16-4 was dissolved in 1 mL N, N-dimethylformamide (DMF). Then, the resulting IMB16-4 solution was dropped into 10 mL 0.1% PVPK30 solution. After stirring at room temperature for 30 min, the mixture was centrifuged to remove the organic solvent. Finally, the precipitation was redispersed with 5 mL 0.1% PVPK30 solution, which was obtained IMB16-4−NPs suspension.

IMB16-4−LNPs and blank liposomes were prepared by thin film hydration method. Briefly, egg yolk lecithin (15 mg) and cholesterol (3 mg) were dissolved in 5 mL N-butanol, and the solution was then evaporated by rotary evaporation to form a homogeneous film at 37 °C. The film was hydrated with 5 mL above IMB16-4−NPs suspension at room temperature and sonicated for 2 min at 150 W. The roughening solution of IMB16-4−LNPs was obtained, and uncoupled IMB16-4 was separated using centrisart (Scilogex, Rocky Hill, Connecticut, USA) and 2.5 mL concentrator (20,000 MWCO) at 15,000 rpm for 20 min at 4 °C. The material trapped in the filter was reconstituted with deionized water and lyophilized with a 6% (*w*/*v*) mixture of sucrose, trehalose, and mannitol for long-term preservation. The final lyophilized powder was stored at 4 °C, which was named IMB16-4−LNPs.

In addition, the film (film containing IMB16-4) was hydrated with distilled water. The blank liposomes suspension (IMB16-4 liposomes suspension) was lyophilized with a 6% (*w*/*v*) mixture of sucrose, trehalose, and mannitol. The final lyophilized powder was stored at 4 °C, which was named blank liposomes (IMB16-4 liposomes).

### 3.3. The Content of IMB16-4−LNPs by HPLC

The encapsulation efficiency (EE) of IMB16-4−LNPs was analyzed on a Shim-pack C18 column (50 × 2.1 mm, 2 μm) at 258 nm using HPLC (LC-2030, Shimadzu, Shimada, Japan). The mobile phase contained 80% methanol and 20% phosphoric acid solutions (pH 2.0). IMB16-4−LNPs suspension was centrifuged for 20 min at 15,000 rpm, and the unentrapped IMB16-4 was removed. The encapsulation efficiency was determined by the following formula:EE (%) = (total amount of drug-unentrapped drug)/total amount of drug × 100IMB16-4−LNPs

### 3.4. Physicochemical Characterization

IMB16-4−LNPs and IMB16-4−NPs were diluted with distilled water, and then their size distribution, polydispersity index, and Zeta potential were measured using the Malvern Zeta sizer Nano ZS.

After being diluted with water, IMB16-4−LNPs and IMB16-4−NPs were dropped on a copper grid, stained with 2% (*w*/*v*) phosphotungstic acid and air-dried at room temperature to investigate the morphology of IMB16−LNPs and IMB16-4−NPs using TEM (JEM1200EX, JEOL, Tokoy, Japan) performed at 80 kV under high vacuum.

FTIR spectra of blank liposomes, IMB16-4, IMB16-4−NPs, and IMB16-4−LNPs were obtained by Fourier transform spectrophotometer (Nicolet IS10, Belmont, MA, USA) with a frequency range of 400 to 4000 cm^−1^.

For DSC analysis, samples were heated from 30 to 300 °C at a heating rate of 10 °C/min under a nitrogen purge of 50 mL/min with a DSC 1 instrument (Mettler Toledo, Sweden).

XRD patterns of IMB16-4, IMB16-4−NPs, and IMB16-4−LNPs were performed from 5° to 40° (diffraction angle 2θ) at a step size of 0.02°with an X-ray diffractometer (Brucker D8 Advance, Karlsruhe, Germany).

### 3.5. Stability Measurements

For long-term storage stability measurements, IMB16-4−LNPs powder was stored at 4 °C, and size, PDI, Zeta potential, and drug content were detected at different time points (10, 20, 30, 40, 50 and 60 days).

### 3.6. In Vitro Drug Release Study

The in vitro drug release from IMB16-4−NPs and IMB16-4−LNPs in PBS (pH 7.4, with 3% SDS) was determined by the dialysis method. Next, 1 mL of both IMB16-4−LNPs and IMB16-4−NPs (0.25 mg/mL of IMB16-4) was placed in dialysis bags (8~12 KMWCO) and submerged in 100 mL PBS (pH 7.4) in thermostatic oscillator (Huamei, Taicang, China) at 37 °C and 100 rpm. Samples were taken at predetermined time intervals and replaced with the same volume of a pre-warmed fresh release medium. IMB16-4 concentrations were assayed by HPLC at a wavelength of 258 nm. The experiment was repeated in triplicates, and data are represented as mean ± SD.

### 3.7. In Vitro Cytotoxicity

The human HSCs line (LX-2) was chosen for in vitro cytotoxicity testing. LX-2 cells were cultured in DMEM/GlutaMAX I (Invitrogen, Carlsbad, CA, USA) with 10% fetal bovine serum and 1% penicillin/streptomycin at 37 °C in 5% CO_2,_ respectively. After being seeded into a 96-well plate and incubated for 24 h, 100 μL of raw IMB16-4, blank liposomes, PVPK30, IMB16-4−NPs, and IMB16-4−LNPs with different concentrations (diluted with DMEM/GlutaMAX I, no FBS) were replaced and incubated for 24 h. Then, 10 μL cell counting kit-8 (CCK8) solutions were incubated together for 2 h. The absorbance was measured at 450 nm by an enzyme microplate reader (BioTek, SYNERGYH1, Winooski, VT, USA).

### 3.8. In Vitro Cellular Uptake

LX-2 cells were grown in a 6-well plate with DMEM/GlutaMAX I, containing 10% fetal bovine serum (FBS). After overnight incubation at 37 °C in 5% CO_2_ atmosphere, the cells were treated with IMB16-4−LNPs or IMB16-4−NPs suspensions at different concentrations of IMB16-4 for 2 h. The cells were washed three times with phosphate-buffered saline (1×X PBS, pH 7.4). Cells were lysed with a 60 µL RIPA buffer containing a protease inhibitor on the ice for 30 min. Total proteins were extracted and quantified using a BCA protein assay kit (Beyotime Biotechnology, Shanghai, China). IMB16-4 concentrations at equal amounts of protein were assayed by HPLC at a wavelength of 258 nm.

### 3.9. The Anti-Fibrotic Effects on LX-2 Cells

LX-2 cells were seeded in a 6-well plate and cultured in DMEM/GlutaMAX I with 10% fetal bovine serum (FBS) in 5% CO_2_ at 37 °C for 24 h. Serum-free culture was replaced for starving cells. After 24 h, cells were treated with TGF-β1 (2 ng/mL) and IMB16-4 (2 µM), IMB16-4−NPs (2 µM), and IMB16-4−LNPs (2 µM) for 24 h. Then, cells were washed with PBS, and protein was extracted with RIPA buffer containing a protease inhibitor. Then mixture was centrifuged at 12,000 rpm at 4 °C for 20 min. The supernatant was collected, and the total protein was determined by a BCA protein assay kit (Beyotime Biotechnology, Shanghai, China). Equivalent amounts of proteins (80 µg) were added into the 10% SDS-PAGE gel. Proteins were transferred onto polyvinylidene difluoride membranes for 100 min, and were blocked with a rapid blocking buffer for 10 min. The membranes were incubated with desired primary antibodies of GAPDH, MMP2, and α-SMA antibodies at 4 °C overnight. After washing with TBST, the membranes were incubated with a horseradish peroxidase (HRP)-conjugated secondary antibody at 1:5000 dilutions for 2 h at room temperature. The protein bands were detected with the Tanon-5200 chemiluminescent imaging system.

### 3.10. Statistical Analysis

One way ANOVA was performed following Dunnett’s test for comparison with the control group. *p* value < 0.05 was considered as statistically significant.

## 4. Conclusions

Herein, we reported a new nanodrug delivery system: IMB16-4−LNPs for IMB16-4. IMB16-4−LNPs possessed an ideal size distribution, preferable biocompatibility, and excellent anti-liver fibrosis effect. IMB16-4−LNPs increased cell uptake compared to IMB16-4−NPs and exhibited higher drug loading compared to liposomes. IMB16-4−LNPs with complementary advantages would be a promising drug delivery in the future.

## Figures and Tables

**Figure 1 molecules-27-03738-f001:**
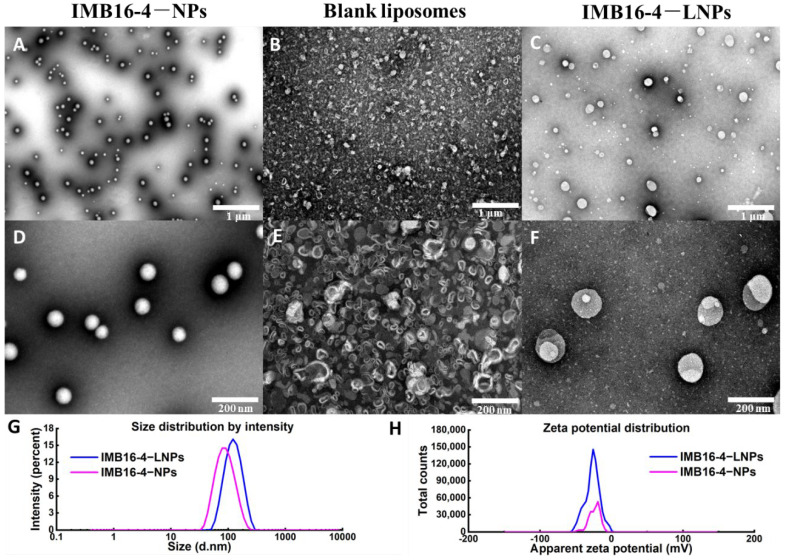
TEM of IMB16-4−LNPs fabricated by coating bilayer lipid membrane to IMB16-4−NPs. (**A**,**D**) IMB16-4−NPs. (**B**,**E**) Blank liposome. (**C**,**F**) IMB16-4−LNPs. (**G**) Particle size and (**H**) Zeta potential of IMB16-4−NPs and IMB16-4−LNPs.

**Figure 2 molecules-27-03738-f002:**
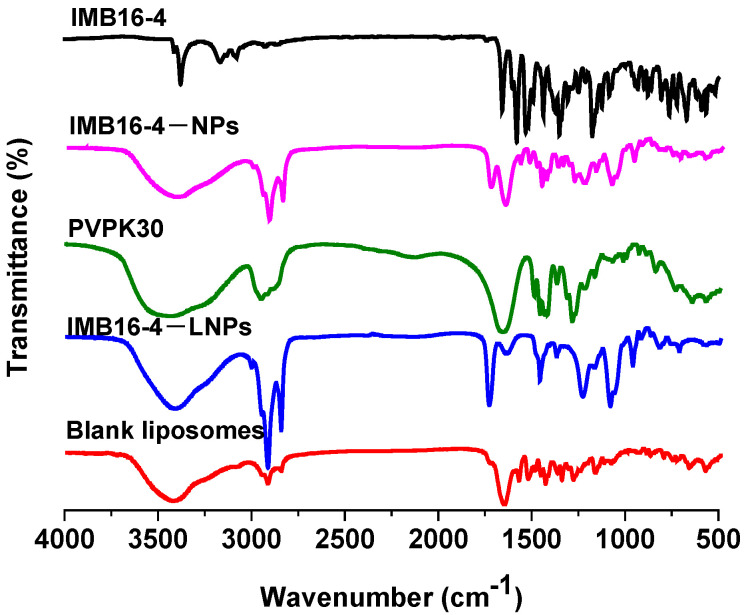
FTIR spectra of blank liposomes, IMB16-4−LNPs, IMB16-4−NPs, and IMB16-4 solid powder with a frequency range of 400 to 4000 cm^−1^.

**Figure 3 molecules-27-03738-f003:**
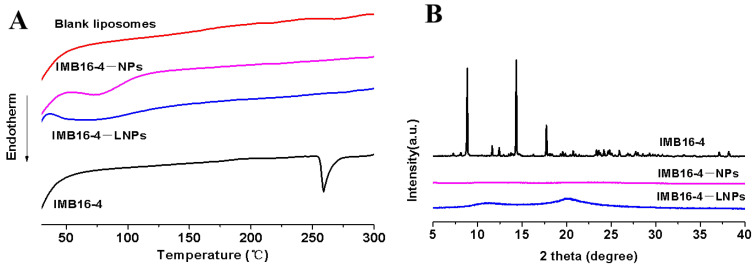
DSC (**A**) and XRD (**B**) analysis of blank liposomes, IMB16-4−NPs, IMB16-4, and IMB16-4−LNPs. For DSC analysis, samples were heated from 30 to 300 °C at a heating rate of 10 °C/min under a nitrogen purge of 50 mL/min. The patterns of XRD were performed from 5° to 40° (diffraction angle 2θ) at a step size of 0.02°.

**Figure 4 molecules-27-03738-f004:**
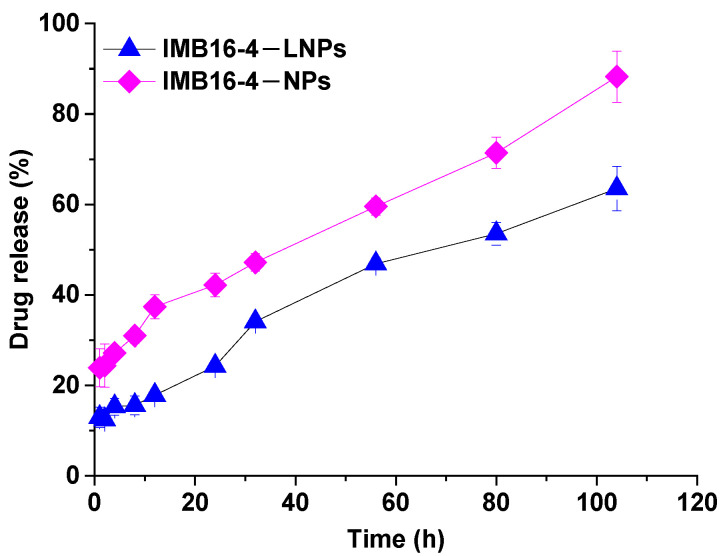
The release of IMB16-4–LNPs and IMB16-4–NPs was carried out by the dialysis method. IMB6-4–NPs were used as control. Then, 1 mL of IMB16-4–LNPs or IMB16-4–NPs containing 0.25 mg IMB16-4 was sealed in dialysis bags and submerged in 50 mL of PBS buffer solution (pH 7.4). Data are presented as mean ± SD (*n* = 3).

**Figure 5 molecules-27-03738-f005:**
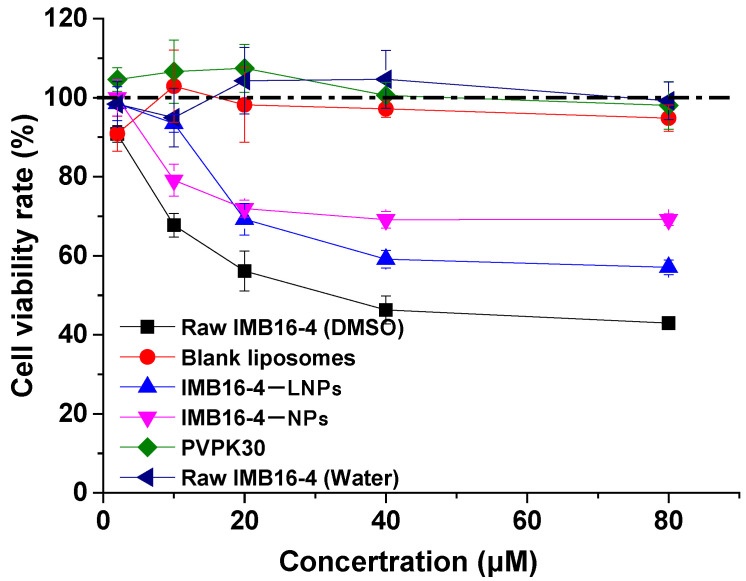
Cell viability of LX-2 cells treated with raw IMB16-4, blank liposomes, and IMB16-4−LNPs for 24 h, respectively. All samples were diluted with DMEM/GlutaMAX I without FBS. The data are presented as mean ± SD of at least triplicate determinations.

**Figure 6 molecules-27-03738-f006:**
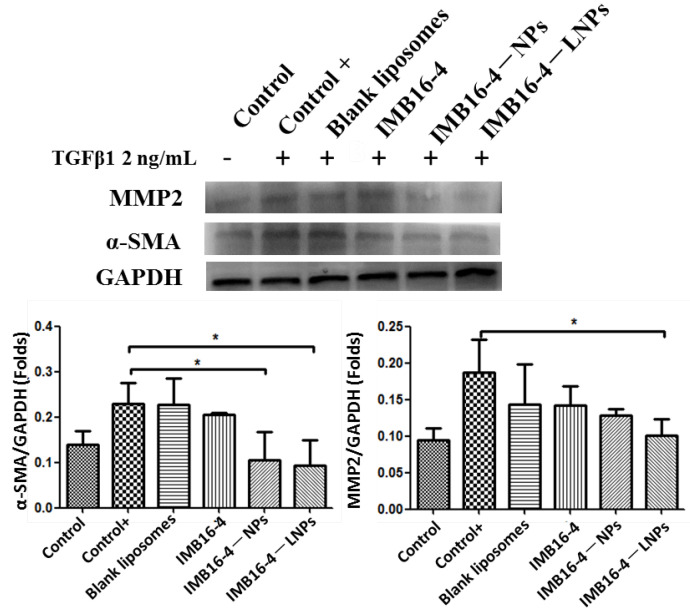
Western blot analysis of MMP2 and α-SMA expression in LX-2 cells. LX-2 cells were treated with 2 ng/mL TGF-β1 with or without 2 μM IMB16-4, IMB16-4–LNPs, and IMB16-4–NPs. The protein levels were normalized against GAPDH. The values are calculated by one way ANOVA following Dunnett’s test for comparison with TGF-β1 treatment group and expressed as the mean ± SD of triplicate independent assays. * *p* < 0.05, significantly different from TGF-β1 treatment group.

**Table 1 molecules-27-03738-t001:** The size, PDI, and Zeta potential of IMB16-4−LNPs and IMB16-4−NPs suspensions.

Sample	Particle Size (nm)	PDI	Zeta Potential (mV)
IMB16-4−NPs	83.8 ± 35.0	0.148	−23.9
IMB16-4−LNPs	119.0 ± 43.1	0.183	−26.6

**Table 2 molecules-27-03738-t002:** The size, PDI, and the content of IMB16-4 of IMB16-4−LNPs lyophilized powder over 60 days. The drug content was measured three times.

Date (Days)	Particle Size (nm)	PDI	ZP	The Content Percentage (%)
0	117.7 ± 52.5	0.267	−26.0	100.0
10	122.3 ± 72.6	0.217	−26.0	104.7
20	121.7 ± 73.8	0.252	−29.9	97.0
30	122.5 ± 73.4	0.218	−25.0	102.2
40	113.1 ± 43.9	0.233	−28.4	95.4
50	119.2 ± 64.7	0.221	−25.0	98.0
60	123.2 ± 45.6	0.241	−26.7	101.7

**Table 3 molecules-27-03738-t003:** Quantitative analysis of cell uptakes. Data are presented as the mean ± SD (*n* = 3).

Concentration (μM)	IMB16-4−NPs (mg/g Prot)	IMB16-4−LNPs (mg/g Prot)
1	0 ± 0	3.60 ± 0.89 **
2	1.48 ± 0.35	6.68 ± 2.85 *

* *p* < 0.05, ** *p* < 0.01, significantly different from IMB16-4−NPs group.

## Data Availability

Not application.

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
