# Peer review of "Established Liposome-Coated IMB16-4 Polymeric Nanoparticles (LNPs) for Increasing Cellular Uptake and Anti-Fibrotic Effects In Vitro"

_molecules, 2022, doi:10.3390/molecules27123738_

Round 1

Reviewer 1 Report

In the current manuscript, Xia Niu and colleagues developed biodegradable liposome-coated polymeric nanoparticles (LNPs) for delivering insoluble cargo MB16-4 for efficient anti-hepatic fibrosis activity. In the present study, the authors performed extensive characterization of hydrophilic cores IMB16-4-NPs and IMB16-4-LNPs for their stability, drug release, and HSCs line (LX-2) cellular uptake. Also, they have validated the cytotoxicity and anti-fibrotic effect in vitro. The article is well written, and the results are presented well. The data is presented with supporting experiments.

However, the authors suggest improving the manuscript by considering the following minor comments.

  1. In the abstract and throughout the article, the first time use of abbreviation should be spelled out clearly. Example: TEM, HPLC, FTIR, LX-2, etc
  2. in Line 35, the author wrote "some types of hepatic fibrosis", this should be clearly explained what types of fibrosis showed hardly any reversal. 
  3. Results and Discussion-The Morphology and Particle Size section showed a ± wider range of size and Zeta potential, Could you explain why and is this system suitable for preclinical experiments?
  4. Most article figures captions need to be expended with the correct information for readers' understanding and are currently too short.
  5. There is no clear information about how many times experiments were repeated, how the statistics were performed, and how many samples were used. At least should be given in the figures. One such example is Figure 6. 
  6. In Figure 4, do you see any statistical significance between the drug release of LNPs and NPs? 
  7. Line no:127, the first time LX-2 abbreviation is used, should be spelled out.
  8. Figure 5, Cell viability assay, have you performed at a higher concentration of cargo to show cellular cytotoxicity of LNPs. In this assay, could you explain why you excluded IMB16-4-NPs? The figure shows A and B, but in the caption hard to guess which graph belongs to what cells. The author should correct the figure captions.
  9. In Figure 6, IMB16-4-LNPs show a very slight differences in reducing expression levels compared to IMB16-4-NPs, could you explain why and is it significant or not? what is the advantage IMB16-4-LNPs in anti-liver fibrosis effects?
  10. What is the status of in vivo absorption and bioavailability, this data would be highly supportive of your cell lines assay and provides solid evidence? This has been performed in your previous IMB16-4-MSN article (https://doi.org/10.3390/molecules26061545).
  11. what is the advantage of this delivery system over your previous IMB16-4-MSN delivery system((https://doi.org/10.3390/molecules26061545)?

Author Response

Response to Reviewer 1 Comments

Point 1:   In the abstract and throughout the article, the first time use of abbreviation should be spelled out clearly. Example: TEM, HPLC, FTIR, LX-2, etc

Response 1: Transmission electron microscope (TEM), high performance liquid chromatography (HPLC), fourier Transform infrared spectroscopy (FTIR) and the human HSCs line (LX-2) were spelled out clearly on the abstract and throughout the article.

Point 2: In Line 35, the author wrote "some types of hepatic fibrosis", this should be clearly explained what types of fibrosis showed hardly any reversal.

Response 2:  According to international standards, the pathological degree of liver fibrosis is divided into 4 stages (S1-S4). In S1 and S2 stage, liver fibrosis can be reversed. In S3 stage, liver fibrosis is not easy to be reversed. In S4 stage, it is virtually impossible to reverse.

Point 3: Results and Discussion-The Morphology and Particle Size section showed a ± wider range of size and Zeta potential, Could you explain why and is this system suitable for preclinical experiments?

Response 3: It's very confusing that particle size section showed narrow particle size distribution in many articles, which was not consisted with the picture. For example, the article  (https://doi.org/10.1038/s41467-019-09852-0) showed average particle size (PS) of 51.72 ± 1.82 nm (mean ± standard error mean, n = 5), and Zeta-potential (ZP) of 18.16 ± 1.00 mV (n = 5). However, the Figure1A (From above reference Figure 1b) hardly showed size distribution of ± 1.82 nm. The reason is that dates of particle size (PS) and Zeta-potential (ZP) were the average of the five results. Every result was tested about 10 times automatically by DLS analyzer.

In our paper, size section showed a ± wider range of size and Zeta potential were from once, which was tested about 10 times automatically by DLS analyzer (Figure 1D and 1E). The average particle size (PS) and Zeta-potential (ZP) of IMB16-4 LNPs were 120.6 ±0.84 nm and -22.8 ± 0.90 mV (mean ± standard error mean, n = 3). These were some of the reasons we showed a ± wider range of size and Zeta potential compared to other papers.

However, particle size of IMB16-4-LNPs was showed on TEM in our paper showed a wider range from obout 50 nm~200 nm. Particle size ranges according to quality standard requirements before preclinical experiments will be strictly controlled,which need further separation and purification.

 Fiure 1 seen in word.

Figure 1. Particle size and Zeta potential. (A) Examples of nanoparticles size distribution from literature. (B) Size distribution of IMB16-4-LNPs and IMB16-4-NPs.(C) Zeta potential distribution of IMB16-4-LNPs and IMB16-4-NPs. (D) Particle size information from one result. (E) Zeta potential information from one result.

Point 4: Most article figures captions need to be expended with the correct information for readers' understanding and are currently too short.

Response 4: Most article figures captions had been expended with the correct information in the revised manuscript.

Point 5: There is no clear information about how many times experiments were repeated, how the statistics were performed, and how many samples were used. At least should be given in the figures. One such example is Figure 6.

Response 5: LX-2 cells were treated with 2 ng/ml TGFβ1 with or without 2 μM IMB16-4, IMB16-4–LNPs and IMB16-4–NPs. The protein levels were normalized against GAPDH. The values are calculated by one way ANOVA following Dunnett’s test for comparing with TGFβ1 treatment group and expressed as the mean ± SD of triplicate independent assays. * p < 0.05, significantly different from TGFβ1 treatment group. The information were added in Figure 6. Others Figure were appropriately added some information.

Point 6: In Figure 4, do you see any statistical significance between the drug release of LNPs and NPs?

Response 6: There is significantly different between the drug release of LNPs and NPs by paired t test. P value was <0.0001.

Point 7: Line no:127, the first time LX-2 abbreviation is used, should be spelled out.

Response 7: Line no:127, the first time LX-2 abbreviation was sustituted with the human HSCs line (LX-2).

Point 8: Figure 5, Cell viability assay, have you performed at a higher concentration of cargo to show cellular cytotoxicity of LNPs. In this assay, could you explain why you excluded IMB16-4-NPs? The figure shows A and B, but in the caption hard to guess which graph belongs to what cells. The author should correct the figure captions.

Response 8:  The toxicity test was poorly designed in our work before. So, cell viability was repeated in these days. We just selected LX-2 cells to conduct the the cell viability further. There was no PBS in the dilution solution in the revised paper. Because there is no FBS during cellular uptake. The groups was divided into raw IMB16-4(DMSO), raw IMB16-4(Water), IMB16-4-LNPs, IMB16-4-NPs, PVPK30 and blank liposome. Raw IMB16-4(DMSO) was dissolved in DMSO, then was diluted with DMEM/GlutaMAX I without FBS. Raw IMB16-4 (Water) was directly suspended in DMEM/GlutaMAX I. Others groups were directly suspended in DMEM/GlutaMAX I.

Raw IMB16-4 was dissolved in DMSO, then was diluted with DMEM/GlutaMAX I without FBS. Obviously, the cell survival rate of IMB16-4-LNPs at 40 μM was higher compared with that of raw IMB16-4. However, raw IMB16-4 was directly suspended in DMEM/GlutaMAX I and showed no cytotoxicity at tested concentration, which might be attributed to low water solubility. The cell survival rate of IMB16-4-LNPs at 40 μM was lower compared with that of IMB16-4-NPs, resulting from increased cellular uptake (Table 3). Interesting, there was no obviously cytotoxicity of IMB16-4-LNPs at the concentration of 2 μM to 10 μM, which enlarged the safe and effective concentration range. The results showed that IMB16-4-LNPs reduced the cytotoxicity and increased the biocompatibility compared to pure IMB16-4. In addition, IMB16-4-LNPs reduced the cytotoxicity at less than the concentration of 10 μM compared to IMB16-4-NPs.

Point 9: In Figure 6, IMB16-4-LNPs show a very slight differences in reducing expression levels compared to IMB16-4-NPs, could you explain why and is it significant or not? what is the advantage IMB16-4-LNPs in anti-liver fibrosis effects?

Response 9: In Figure 6, IMB16-4-LNPs show hardly significance in reducing expression levels compared to IMB16-4-NPs. The reason was that the expression levels of IMB16-4-LNPs or IMB16-4-NPs was both more approaching control-. LX-2 cell in control- group were not treated by TGFβ1 and cargo.

Point 10: What is the status of in vivo absorption and bioavailability, this data would be highly supportive of your cell lines assay and provides solid evidence? This has been performed in your previous IMB16-4-MSN article (https://doi.org/10.3390/molecules26061545).

Response 10: IMB16-4-LNPs was used for intravenous injection and we did not focus on the absorption problem in vivo. We constructed the IMB16-4-LNPs for intending to increase the passive liver target and hepatocyte absorption.

Point 11:  what is the advantage of this delivery system over your previous IMB16-4-MSN delivery system((https://doi.org/10.3390/molecules26061545)?

Response 11: MSN as inorganic carrier showed controversial accumulates toxicity in body. IMB16-4-LNPs showed high biocompatibility, low immunogenicity and biodegradable nanocarry.

Reviewer 2 Report

I read with much interest the article entitled "Established Liposome-Coated IMB16-4 Polymeric Nanoparticles (LNPs) for Increasing Cellular Uptake and Anti-fibrotic Effects in Vitro" by Xia Niu, Yanan Meng, Yucheng Wang and Guiling Li.

     The manuscript is well structured and contained up-to-date references. I am convinced that it will be of potential interest to the science community. According to my opinion, the manuscript can be considered for publication.

Major concerns:

2. Subsection 2.1

The results from the application of the DLS technique should be shown also in the figure, e.g. size distribution by number or intensity.

2. The cytotoxicity determination (2.7 and 3.6 subsections)

As a chemist, I only had to handle MTT and ATP assays to predict the potency of cytotoxic agents in various human cancer cell lines.

Therefore, I do not understand why absorbance was included in the methodology part? Was the cell viability rate determined using an optical microscope? Please explain.

Minor concerns:

1. Line 75 - the abbreviation EE for encapsulation efficiency was not formally introduced in this article.

2. Line 93 - bland?

3. Line 115 - what does SDS stand for?

3. Line 169 - "...were obtained as previously reported." - such a statement forces the citation of the literature.

 4. Line 180 - please specify the company from which centrisart was purchased.

 4. Line 234 - what does CCK8 stand for?

 5. Line 240 - what is the meaning of 100?

Author Response

Response to Reviewer 2 Comments

Major concerns:

Point 1:    Subsection 2.1

The results from the application of the DLS technique should be shown also in the figure, e.g. size distribution by number or intensity.

Response 1: The figure of size distribution by intensity was added in Figure 1.

Point 2: The cytotoxicity determination (2.7 and 3.6 subsections)

As a chemist, I only had to handle MTT and ATP assays to predict the potency of cytotoxic agents in various human cancer cell lines.

Therefore, I do not understand why absorbance was included in the methodology part? Was the cell viability rate determined using an opt`ical microscope? Please explain.

Response 2: CCK-8 is more sensitive than MTT,XTT,MTS or WST-1.

CCK-8 allows very convenient assays by utilizing Dojindo’s highly water-soluble tetrazolium salt. WST-8[2-(2-methoxy-4-nitrophenyl)-3-(4-nitrophenyl)-5-(2, 4-disulfonic acid benzene) -2H – tetrazolium, monosodium salt] produces a water-soluble formazan dye upon reduction in the presence of an electron mediator and gives an orange colored product (formazan), which is soluble in the tissue culture medium. The amount of the formazan dye generated by dehydrogenases in cells is directly proportional to the number of living cells. Microplate reader (BioTek, SYNERGYH1, America)was used to measure the light absorption value at 450 nm, which could indirectly reflect the number of living cells. This method has been widely used in the activity detection of some bioactive factors, large-scale screening of anti-tumor drugs, cell proliferation test, cytotoxicity test and drug sensitivity test.

Minor concerns:

Point 1: Line 75 - the abbreviation EE for encapsulation efficiency was not formally introduced in this article.

Response 1: Entrapment efficiency (EE) was added in line74.

Point 2: Line 93 - bland?

Response 2: It was spelling mistake. We have changed into blank in line 93.

Point 3: Line 115 - what does SDS stand for?

Response 3: Sodium dodecyl sulfate stands for SDS. Sodium dodecyl sulfate was added in line 115.

Point 4: Line 169 - "...were obtained as previously reported." - such a statement forces the citation of the literature.

Response 4: The literature was cited in line169.

Point 5: Line 180 - please specify the company from which centrisart was purchased.

Response 5: The centrisart of brand was Scilogex in Amarica. We added in line 180.

Point 6: Line 234 - what does CCK8 stand for?

Response 6:  Cell counting kit-8 was abbreviated CCK8. Cell counting kit-8 was added in Line 234.

Point 7: Line 240 - what is the meaning of 100?

Response 7: 100 in line 240 was inserted by mistake. We've deleted it.

Reviewer 3 Report

The topic is essential. Research is done systematically and results are presented professionally. However, I would recommend the authors elaborate on the conclusion a bit more including some of the bold aspects of the results.

Author Response

Response to Reviewer 3 Comments

The topic is essential. Research is done systematically and results are presented professionally. However, I would recommend the authors elaborate on the conclusion a bit more including some of the bold aspects of the results.

Response :We have elaborated the conclusion .

Round 2

Reviewer 1 Report

The authors addressed most of my suggestions and I am happy with the answers. Therefore I do not have further comments.